# Semi-Supervised Learning via Clustering Representation Space

## Abstract

We proposed a novel loss function that combines supervised learning with clustering in deep neural networks. Taking advantage of the data distribution and the existence of some labeled data, we construct a meaningful latent space. Our loss function consists of three parts, the quality of the clustering result, the margin between clusters, and the classification error of labeled instances. Our proposed model is trained to minimize our loss function, avoiding the need for pre-training or additional networks. This guides our network to classify labeled samples correctly while able to find good clusters simultaneously. We applied our proposed method on MNIST, USPS, ETH-80, and COIL-100; the comparison results confirm our model's outstanding performance over semi-supervised learning.

## 1 Introduction

Labeling data is expensive. Thus, it is often hard for us to get enough labeled samples. Thus, semi-supervised learning (Chapelle et al., 2009) becomes a serious issue. People try to get good performance with limited labeled data and a large amount of unlabeled data. When having a limited amount of labeled samples, extracting information from unlabeled data has played an important role for semi-supervised learning. In general, we often applied unlabeled information as auxiliary tools, such as pre-training (Hinton and Salakhutdinov, 2006) or recursive picking confidence data from unlabeled samples with supervised learning(Zhu, 2005). However, we notice that when we counter the issue such as Two Half-moon, double circle, or other more complex distribution problem, these methods are lack of considering the spatial distribution information provided by unlabeled samples.

In this paper, we aimed to guide our model to extract the spatial distribution information from unlabeled data. We proposed a new approach for semi-supervised learning by adding our loss function term for our target embedding latent space. Within our proposed model, the neural network can now learn correctness and spatial distribution information from labeled and unlabeled samples simultaneously. This provides our feed-forward neural network to have more opportunity passing through the sparse margin between clusters, and elevate the performance of the classifier, see more details in Sec. 3. Moreover, it is worth noting that our proposed model does not rely on any additional neural networks, which is suitable for any task and is highly compatible with different semi-supervised learning algorithms.

In short, the characteristics of our proposed model are as follows:
**Intuitive** The idea of correctness and spatial distribution came up with the characteristics of supervised and unsupervised learning straightly, which is intuitive.
**Compatibility** Our method does not rely on any additional neural networks but only adding new loss term. Our approach is easy to change into any existing feed-forward neural networks.
**Extensible** We designed our approach by the notion of defining an evaluation for spatial distribution, which can replace by any other methods in future researches.

## 2 Related work

In recent years, the neural network plays an essential role in various tasks; more specifically, they are highly applied in the tasks of image classification. Since then, semi-supervised learning (Weston et al., 2012; Lee, 2013) for image classification has become a vital issue. First of all, some works

succeeded by proposing regularization methods for NNs (Bishop, 1995; Srivastava et al., 2014). They regularize the input and hidden layers of their models by applying random permutations. This can smooth the input-output relation and further get improvements in semi-supervised learning.

Generative adversarial networks (GAN) (Goodfellow et al., 2014) are popular research for the neural network, several models (Salimans et al., 2016; Dumoulin et al., 2016; Dai et al., 2017) had gone more in-depth researches with GAN for semi-supervised learning. They showed remarkable results, especially on image classification problems. In 2014, Kingma et al. proposed Deep generative models (Kingma et al., 2014), applying a variational-autoencoder based generative model to semi-supervised learning. They showed good results on image classification for semi-supervised learning and became the benchmark of several datasets. However, in practice, these models require more careful tuning for parameters. Also, they usually require more neural network structures and computation resources.

Moreover, Rasmus et al. (Rasmus et al., 2015) propose semi-supervised with Ladder Networks, a model structure with an autoencoder. This work is similar to denoising autoencoders and applied to every layer. It is impressive that they have got a vast improvement compared with Deep Generative models (Kingma et al., 2014). Shortly after, Miyato ey al. (Miyato et al., 2018) also achieve competitive results on the benchmark data sets with a regularization term. They guide their model to minimize the change of the input and output of the network, which does not require labeled information and able to use unlabeled samples for their regularization term.

Labeled propagation proposed by (Zhu and Ghahramani, 2002) is also a family of methods for semi-supervised learning. By smoothing the model around the input data points, the model can extrapolate the labels of unlabeled samples. Similar to this idea, several works (Laine and Aila, 2016; Sajjadi et al., 2016) had succeeded by using the random image augmentation. They try to improve the generation performance of image classification for semi-supervised learning.

## 3 PROPOSED MODEL

For semi-supervised learning, we assume that in some particular space, samples in the same category should be in the same cluster. Following this assumption, once we can distinguish different clusters properly, and guide our network to find the decision boundary with sparser region between clusters.

In this section, we introduce an end-to-end learning method by adding our loss functions. First of all, we tried to guide our network to learn a good mapping from the original input space to the embedding latent space; see Sec. 3.1. We next defined loss functions and tried to cluster the samples that should be in the same categories together in the embedding latent space; see Sec. 3.2. Moreover, similar to some supervised learning works (Xu et al., 2005; Rennie and Srebro, 2005; Srebro et al., 2005), we aim to maximize the margin between different clusters to separate them well. Note that since we are lack of labeled samples, we maximize the margin within temporary clustering results for our data, instead of using the ground truth of labeled data, see more in Sec. 3.3. Overall, we named our proposed model as **Maximum Cluster Margin Classifier**, referred to as **MCMC**.

### 3.1 EMBEDDING LATENT SPACE

In general, measuring a clustering result is very subjective. However, to deal with various kinds of distribution, we avoid evaluating them directly by their original input space. Instead, we pull out a layer from the neural network and set it as the embedding latent space. Next, we evaluate whether the quality of the embedding latent space. In our proposed model, we try to define a measurement for the embedding latent space to satisfy our assumptions. This guides the previous layers to learn about a good mapping from the original input space to a well-distributed embedding latent space. To strengthen the efficiency of the embedding latent space, we add a simple classifier that is fully connected to the embedding latent space.

### 3.2 DB LOSS

#### 3.2.1 DAVIES-BOULDIN INDEX

As Davies-Bouldin index (Davies and Bouldin, 1979) proposed, given a dataset with $N$ clusters, for every cluster $C_i$, we compute $S_i$ as a measure of scattering with the cluster, which is defined as

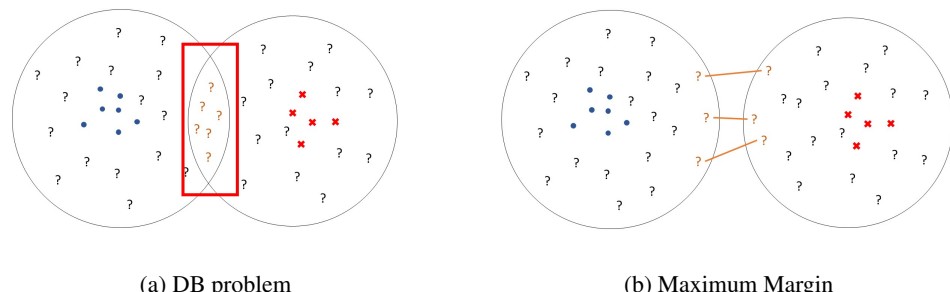

(a) DB problem                                     (b) Maximum Margin

Figure 1: (a) Since DB loss $\mathcal{L}_{DB}$ consider about the centroid of cluster only, this leads the problem of overlapping between different clusters. (b) We compute Maximum Margin loss $\mathcal{L}_{MM}$ by maximizing these $k$ pairs single link.

follows:

$$S_i = (\frac{1}{T_i} \sum_{j=1}^{T_i} |\mathbf{X}_i^j - A_i|^p)^{\frac{1}{p}} \tag{1}$$

where $\mathbf{X}_i^j$ is the $jth$ sample of cluster $C_i$, $A_i$ is the centroid of cluster $C_i$, $T_i$ is the size of $C_i$ and $p$ is usually set as 2. To measure the distance between different centroid of clusters $C_i$ and $C_j$, we compute $M_{i,j}$, which is defined as follows:

$$M_{i,j} = ||A_i - A_j||_p = (\sum_{k=1}^{n} |a_{k,i} - a_{k,j}|^p)^{\frac{1}{p}} \tag{2}$$

where $a_{k,i}$ is the $i$th element of $A_i$, and $p$ is usually set as 2. Finally, we combine $S_i$ with $M_{i,j}$, which is defined as follows:

$$R_{i,j} = \frac{S_i + S_j}{M_{i,j}}, \quad D_i \equiv \max_{i \neq j} R_{i,j}, \quad DB \equiv \frac{1}{N} \sum_{i=1}^{N} D_i. \tag{3}$$

### 3.2.2 DB Loss

Thanks to Davies-Bouldin index Sec. 3.2.1 (Davies and Bouldin, 1979), we now have a measurement for evaluating the clustering result in embedding latent space. Eq. (1) provides us the evaluation of whether the samples in a single cluster are close enough to the centroid. Simultaneously, Eq. (2) separates the centroid of different clusters to become as far as possible. Moreover, Eq. (3) restricts every single cluster could gather together while different cluster has a long distance between their centroid. We defined our DB loss $\mathcal{L}_{DB}$ as follows:
Given points $\mathbf{X}$ with temporary predictions $\hat{y}$, then

$$\mathcal{L}_{DB} \equiv DB \tag{4}$$

where $DB$ can be computed in Sec. 3.2.1.

Till now, DB loss $\mathcal{L}_{DB}$ provides us the capability to guide the samples in the same category gather in the same cluster. However, when we look into Davies-Bouldin index (Davies and Bouldin, 1979), the measurement considers only about the centroid of each cluster. This causes the issue that different clusters may be overlap with each other. That is, the margin between different clusters has vanished, see Fig.3. For this reason, we came up with Maximum Margin loss, see Sec. 3.3.

### 3.3 Maximum Margin Loss

As we mentioned at the start of Sec. 3, we need to maximize the margin between different clusters to separate them. We hence came up with Maximum Margin Loss $\mathcal{L}_{MM}$. To push each cluster as far as we can, we compute the sum of $k$ pairs single link (Gower and Ross, 1969) between different clusters. When maximizing those pairs of single link, it also implies maximizing the margin between

different clusters. We define the Maximum Margin loss $\mathcal{L}_{MM}$ as follows:

Given clusters $C_1, C_2, ..., C_N$, we let $\mathcal{V}_{i,j}^\ell$ be the $\ell^{th}$ pair single link between $C_i$ and $C_j$. Also, we define $\sum_{\ell=1}^{k} \frac{1}{V_{i,j}^\ell}$ to be $k$ pairs single link distance$_{i,j}$. Finally, we have

$$\mathcal{L}_{MM} = \sum_{i \neq j, i < j}^{N} (k \, pairs \, single \, link \, distance_{i,j}), \tag{5}$$

where $k$ is the number of pairs between different clusters, and $N$ is the number of the clusters.

Maximum Margin loss $\mathcal{L}_{MM}$ provides the information of k-pair single link distance between different clusters. This loss can guide the model to ameliorate the overlap problem and maximize the margin between different clusters. Fig. 1(b) shows the main idea of Maximum Margin loss $\mathcal{L}_{MM}$, which helps our network avoid the problem illustrated in Fig. 1(a). Overall, by combining DB loss $\mathcal{L}_{DB}$ and Maximum Margin loss $\mathcal{L}_{MM}$, we can find our ideal evaluation for clustering results in embedding latent space. Our network is now able to gather the samples that should be in the same categories and separate different clusters as far as possible.

### 3.4 TOTAL LOSS

In this section, we took a more in-depth exploration of how we deal with labeled data and unlabeled data for our total loss separately. For labeled data, we applied loss with cross-entropy (CE) (De Boer et al., 2005), which is widely used for classification tasks in supervised learning. Note that Cross-entropy (CE) (De Boer et al., 2005) is available to change into other losses in supervised learning for personal use. Though labeled data might have a small number of samples, they still play an essential role in guiding the network learning the right predictions. Moreover, they can also provide the neural network to learn a rough decision boundary in the early iterations. This leads our network to get a good start and modify the boundary into better results. As we mentioned in Sec. 3.2.2 and Sec. 3.3, we need DB loss $\mathcal{L}_{DB}$ to guide the samples in the same cluster to group together. At the same time, Maximum Margin loss $\mathcal{L}_{MM}$ pushes the different clusters as far as possible, i.e., maximize the margin between different clusters. We define the total loss $\mathcal{L}_{Total}$ as follows:

$$\mathcal{L}_{Total} = \quad \mathcal{L}_{DB} + \mathcal{L}_{CE} + \mathcal{L}_{MM}. \tag{6}$$

### 3.5 COMPUTING FLOW

For our network training, since we need several points to compute our loss in a single iteration, we assume every batch has enough samples. During our computations, we separate DB loss $\mathcal{L}_{DB}$ into two parts. First of all, we sampled some data as a batch from all training data, including labeled data and unlabeled data. We compute our DB loss for all training data $\mathcal{L}_{DB}^T$ by the embedding latent space and temporary predictions. Next, we compute another DB loss for labeled data $\mathcal{L}_{DB}^L$ with the embedding latent space and their ground truth labels. We found that adding this loss can speed up training and make our network more stable. It is reasonable since we use ground truth instead of temporary predictions for computing DB loss for labeled data $\mathcal{L}_{DB}^L$. This can guarantee that labeled samples in the same categories gather in the same cluster within their ground truth. Hence, it plays an important role in finding a well-distributed embedding latent space that satisfies our assumptions mentioned in the start of Sec. 3.

During our training, we give two different batch sizes for $\mathcal{L}_{DB}^L$ and $\mathcal{L}_{DB}^T$. Please note that batch for $\mathcal{L}_{DB}^L$ can always get the labeled samples. Another side, batch for $\mathcal{L}_{DB}^T$ sampled data from all training data, has the possibility to get samples with no labeled data for a single batch. This fact also shows the importance of $\mathcal{L}_{DB}^L$ to speed up gathering the samples in the same category together. Overall, the final DB loss $\mathcal{L}_{DB}$ is defined as follows:

$$\mathcal{L}_{DB} = \mathcal{L}_{DB}^L + \mathcal{L}_{DB}^T. \tag{7}$$

Moreover, to maximize the margin between different clusters, we compute Maximum Margin loss $\mathcal{L}_{MM}$ by the same process as DB loss $\mathcal{L}_{DB}^T$ for training data. We now divide computing flow into

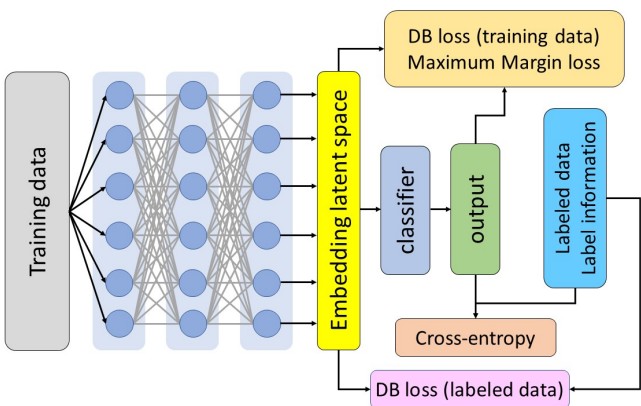

Figure 2: Computing flow for our losses in MCMC. We compute cross-entropy by the output of temporary predictions and the ground truth of labeled data, $\mathcal{L}_{DB}^{L}$ by the embedding latent space and the ground truth of labeled data, $\mathcal{L}_{DB}^{T}$ and $\mathcal{L}_{MM}$ by the embedding latent space and the output of temporary predictions. Finally, we sum up our total loss for MCMC.

---

**Algorithm 1** Training for Maximum Cluster Margin Classifier

---

**Input:** Network $z(\cdot; \theta_z)$; classifier $g(\cdot; \theta_g)$; data $D_L, D_U, D_T = D_L \cup D_U$; batch sizes $N_L, N_T$;
  learning rate $\alpha$
**Output:** Learnt network parameters $z(\cdot; \theta_z), g(\cdot; \theta_g)$
 1: Initialize parameters $z(\cdot; \theta_z), g(\cdot; \theta_g)$
 2: **repeat**
 3:   $(\mathbf{X}_L, y) \leftarrow N_L$ samples from $D_L$;
 4:   $(\mathbf{X}_T) \leftarrow N_T$ samples from $D_T$;
 5:   $\mathbf{Z}_L \leftarrow z(\mathbf{X}_L; \theta_z); \mathbf{Z}_T \leftarrow z(\mathbf{X}_T; \theta_z)$;
 6:   $y_L \leftarrow g(\mathbf{Z}_L; \theta_g)$;
 7:   $y_T \leftarrow g(\mathbf{Z}_T; \theta_g)$; *%Temporary predictions for training data.*
 8:   $L_{CE} \leftarrow CrossEntropy(y, y_L)$;
 9:   $L_{DB}^{L} \leftarrow DBI(\mathbf{Z}_L, y); L_{DB}^{T} \leftarrow DBI(\mathbf{Z}_T, y_T)$;
10:   $L_{MM} \leftarrow MaximumMarginLoss(\mathbf{Z}_T, y_T)$;
11:   $L_{Total} \leftarrow L_{CE} + L_{DB}^{L} + L_{DB}^{T} + L_{MM}$;
12:   $\theta_z \leftarrow \theta_z - \alpha \frac{\partial L_{Total}}{\partial \theta_z}; \theta_g \leftarrow \theta_g - \alpha \frac{\partial L_{CE}}{\partial \theta_g}$;
13: **until** stopping criterion is $true$

---

labeled data and training data, which can also imply learning the correctness and modifying a better decision boundary separately. Fig. 2 shows our entire computing flow for each loss and Algorithm 1 introduces computing details for our training. For the convenience during our experiments, we set $k = 10$ for $\mathcal{L}_{MM}$ without fine-tuning.

## 4   TOY PROBLEMS

To justify our theory, we first conduct a study with two classic experiments, Two Half-moon, and Triple Circle problem. In our experiments, we generate samples with Scikit-learn (Pedregosa et al., 2011) and randomly select labeled samples to reproduce the situational simulation of semi-supervised learning. We present the transformation process of the decision boundary. Also, we show the failure case for supervised learning to demonstrate the improvements between MCMC and supervised learning. Finally, we take a closer observation of our proposed model.

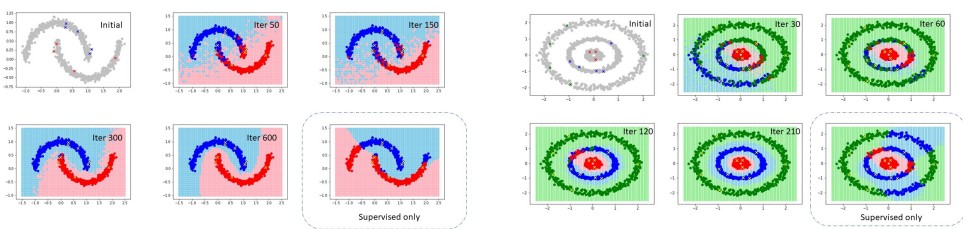

(a) Two Half-moon experiment          (b) Triple Circle experiment

Figure 3: Main figures show the initial labeled samples and the transformation process of the decision boundary for MCMC. The dashed box shows the failure case of supervised learning with labeled samples. (a) Two Half-moon experiment. (b) Triple Circle experiment.

**Two Half-moon problem** For Two Half-moon problem, we randomly sampled 1000 points with Scikit-learn (Pedregosa et al., 2011) and draw 9 points of them as labeled data. As Fig. 3(a) shows, when adding our loss functions, the decision boundary was modified to a better result with considering the spatial distribution information. Moreover, we notice that unlike supervised learning, when our proposed model is trying to separate two clusters, it tries to maintain the correctness of the labeled red points simultaneously.

**Triple Circle problem** We applied our proposed model to Triple Circle problem. We sampled 1500 points with Scikit-learn (Pedregosa et al., 2011) randomly and draw 13 points as labeled data. Fig. 3(b) shows our impressive improvements when considering the spatial distribution with unlabeled samples. On the other side, the result of supervised learning gets an unfortunate result though all labeled data are actually in the right categories.

We notice that for multi categories problem, it is still crucial to learn a rough decision boundary in the early iterations. Overall, we fit our original hypothesis in this experiment. We show that our proposed loss function can provide the spatial distribution information from unlabeled data and adjust the decision boundary without losing the correctness during training. By decreasing loss within more iterations, the decision boundary can maximize the margin between different clusters. This is precisely in line with our original expectations and is able to extract spatial distribution information from unlabeled samples for semi-supervised learning. These results not only strengthen our confidence for our loss function but also shows impressive works during multi categories problem.

## 5 Evaluation on Real data

We evaluate our model on four real data sets, MNIST, USPS, ETH-80, and COIL-100. We reproduce the experiments of the MNIST data set followed by Kingma et al. (Kingma et al., 2014) and compare them with different works. As for USPS, ETH-80, and COIL-100 data sets, we follow Zhang et al.(Zhang et al., 2020) and summarize the result of performance comparison. For these experiments, we implement our model within PyTorch (Paszke et al., 2017). Thanks for the tool; it helped us done differentiation automatically. Our open source will release in Github after our paper is accepted.

### 5.1 Results for the MNIST dataset

MNIST dataset is consists of handwriting digit images with image size 1x28x28, including 60000 training data and 10000 testing data. Following the works of Kingma et al. (Kingma et al., 2014), we use different numbers of samples for $D_L$ from the training set. Note that all classes are balanced, i.e., each class has the same number of labeled samples. Since our network structure is easy to change into any neural network, in this task, we replace our network into a convolution neural network. We followed the ConvPool-CNN-C structure from Springenberg et al. (Springenberg et al., 2014). We add our embedding space before the last layer, which is set as 300 and fully connected to 10 neural (10 classes). For training, we used ADAM (Kingma and Ba, 2014) as our optimizer, with a learning

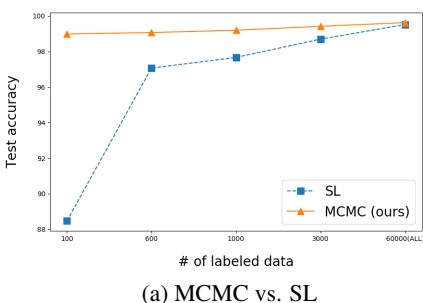
(a) MCMC vs. SL

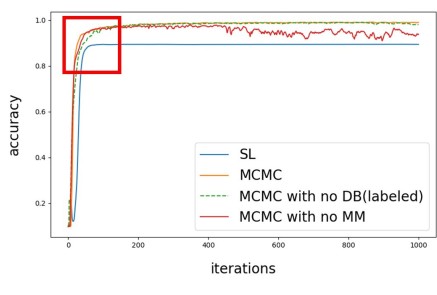
(b) Ablation study

Figure 4: (a) Test accuracy with different amounts of labeled data on MNIST. Figure shows the result of supervised learning (SL) and our proposed model 'MCMC'. (b) The transitions of accuracy during different methods on MNIST with 100 labeled data. We tested with four methods, supervised learning (SL), MCMC, MCMC without DB loss for labeled data, and MCMC without Maximum Margin loss. The red box shows the differs in the early iterations between different methods.

rate of 0.001, and exponentially decayed the rate during training. We use 50 and 300 for the batch size of labeled data and training data separably. Note that we did not build validation data but fixed our training epochs as 30 epochs for every experiment. Also, we used $k = 10$ for $\mathcal{L}_{MM}$ and did not fine-tune any parameters or structures. We repeated the experiments ten times with different random seeds for labeled data.

We ran some experiments with different settings on MNIST for 100 labeled samples. As Fig. 4(a) shows, supervised learning within a small amount of labeled data shows poor test accuracy and elevates the performance as having more labeled data. Compared with SL, MCMC shows significant improvements in the performance of test accuracy. Fig. 4(b) provides the ablation study results on MNIST with 100 labeled data. In Fig. 4(b), MCMC shows the strongest stability with high accuracy, and supervised learning performs the worst. For MCMC without DB loss for labeled data, the model takes more iterations to achieve better accuracy. As we introduced in Sec. 3.5, this loss is important for the network to learn faster during training. For MCMC without Maximum Margin loss, we found the instability for the network during the training. Without Maximum Margin loss, the embedding latent space will occur the overlap problem that showed in Fig. 1(a). This causes the failure case for semi-supervised learning. As a result, it is crucial for us to add both Maximum Margin loss and DB loss for labeled data, making our model become more stable and further increase the accuracy.

Table 1 summarizes the results of MNIST dataset and shows the reports of the mean and standard deviation with different amounts of labeled data. The mean and standard deviation of error rate both decrease as we add more labeled data. This also implies that labeled samples stand a critical role when training model once again. In Table 1, our proposed method shows remarkable results and outperforms with other methods.

Table 1: Test performance on MNIST dataset with the different number of labeled data.

| Test error % with # of used labels | 100 | 1000 | ALL |
|---|---|---|---|
| TSVM (Collobert et al., 2006) | 16.81 | 5.38 | 1.40 |
| MTC (Rifai et al., 2011) | 12.03 | 3.64 | 0.81 |
| AtlasRBF (Pitelis et al., 2014) | 8.10 ($\pm$0.95) | 3.68 ($\pm$0.12) | 1.31 |
| DGM (M1+M2) (Kingma et al., 2014) | 3.33 ($\pm$0.14) | 2.40 ($\pm$0.02) | 0.96 |
| CatGAN (Springenberg, 2015) | 1.91 ($\pm$0.10) | 1.73 ($\pm$0.18) | 0.91 |
| Conv-CatGAN (Springenberg, 2015) | 1.39 ($\pm$0.28) | - | 0.48 |
| VAT (Miyato et al., 2018) | 1.36 ($\pm$0.03) | 1.27 ($\pm$0.11) | 0.64 ($\pm$0.05) |
| Ladder, full (Rasmus et al., 2015) | 1.06 ($\pm$0.37) | 0.84 ($\pm$0.08) | 0.57 ($\pm$0.02) |
| Ladder, CNN-$\Gamma$ (Rasmus et al., 2015) | 0.89 ($\pm$0.50) | - | - |
| MarginGAN (Dong and Lin, 2019) | 3.53 ($\pm$0.57) | 2.87($\pm$0.71) | - |
| FlowGMM-cons (Izmailov et al., 2019) | 1.8 | 1.0 | - |
| MCMC (ours) | **1.01** ($\pm$0.13) | **0.80** ($\pm$0.07) | **0.37** ($\pm$0.05) |

## 5.2 Results for the USPS, ETH-80, COIL-100 dataset

To evaluate our method in more real problems, we took experiments on USPS, ETH-80, and COIL-100. Followed by Zhang et al. (Zhang et al., 2020), we took 12 different tests with various settings for each data set, and compare with other works.

**USPS data set**    USPS (Hull, 1994) is a database of handwritten digits, includes 11000 images with image size 1x16x16. We took experiments with different amounts of labeled samples per class. The number of the labeled digit is set to 5, 10, ..., 60, corresponds to 12 experiments.

**ETH-80 data set**    ETH-80 (Leibe and Schiele, 2003) contains 3280 object images with 8 broad categories and ten subcategories for each category. During our preprocessing, we resized every image into 3x32x32. We took experiments with different amounts of labeled samples per class. The number of the labeled digit is set to 10, 25,..., 175, corresponds to 12 experiments.

**COIL-100 data set**    COIL-100 (Nene et al., 1996) includes 7200 images with 100 different objects. During our preprocessing, we resized every image into 3x32x32. We took experiments with different amounts of labeled samples per class. The number of the labeled digit is set to 2, 4,..., 24, corresponds to 12 experiments.

Table 2 shows the statistic of 12 experiments for each data set and the comparison with other works. Different from the concept of label propagation or similar works, we do not define the similarity between samples. We assume that the similar samples are also similar in our embedding space, and gather them together by optimizing our loss function. This force the dense samples converge into the same cluster in our embedding space and make our proposed method outperformed with other works.

Table 2: Performance comparison of USPS, ETH-80 and COIL-100

|  | USPS | ETH-80 | COIL-100 |
|---|---|---|---|
|  | Mean$\pm STD$ | Mean$\pm STD$ | Mean$\pm STD$ |
| GFHF (Zhu et al., 2003) | 72.57 ($\pm$8.72) | 71.86 ($\pm$5.50) | 71.86 ($\pm$5.50) |
| LLGC (Zhou et al., 2004) | 85.34 ($\pm$6.52) | 81.62 ($\pm$2.71) | 86.36 ($\pm$3.78) |
| LNP (Wang and Zhang, 2007) | 86.57 ($\pm$6.25) | 80.86 ($\pm$2.68) | 86.97 ($\pm$3.49) |
| SLP (Nie et al., 2010) | 85.89 ($\pm$5.25) | 80.30 ($\pm$2.89) | 87.36 ($\pm$3.38) |
| PN-LP (Zoidi et al., 2016) | 80.23 ($\pm$6.24) | 72.41 ($\pm$4.42) | 87.18 ($\pm$3.83) |
| CD-LNP (Zhang et al., 2015a) | 73.47 ($\pm$9.22) | 70.11 ($\pm$4.83) | 82.91 ($\pm$4.81) |
| SparseNP (Zhang et al., 2015c) | 89.93 ($\pm$2.42) | 82.81 ($\pm$2.02) | 76.36 ($\pm$5.34) |
| ProjLP (Zhang et al., 2015b) | 89.68 ($\pm$3.22) | 82.39 ($\pm$1.95) | 90.41 ($\pm$3.10) |
| AdaptiveNP (Jia et al., 2016) | 82.37 ($\pm$5.99) | 73.29 ($\pm$4.34) | 78.42 ($\pm$4.54) |
| ALP-TMR (Zhang et al., 2020) | 91.12 ($\pm$3.04) | 82.26 ($\pm$1.95) | 91.51 ($\pm$4.13) |
| MCMC (ours) | **94.28** ($\pm$2.38) | **93.38** ($\pm$5.91) | **94.74** ($\pm$3.18) |

## 6 Conclusion

In this paper, we proposed a new approach, which we called Maximum Cluster Margin classifier (MCMC), for semi-supervised learning. The key idea is that we define a novel loss function for clustering results. In particular, the quality of embedding latent space plays an important role for semi-supervised learning. In our embedding latent space, we guide our network to gather the samples which should be in the same categories. Simultaneously, pushing these clusters as far as possible, and find a better boundary passing through the sparse region. We showed the indispensability for MCMC to have high stability during training for our losses. Also, we compared with other works and showed outstanding improvements for semi-supervised learning on these data sets.

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
