# OpenReview forum: "Semi-Supervised Learning via Clustering Representation Space"
_ICLR.cc/2021/Conference — Reject_

### Official Review · AnonReviewer1 · 2020-10-27
**The method is correct, and is simple and straightforward, which is not necessarily bad. However, the paper has several serious issues with the empirical evaluations. The paper is not at the bar to be accepted.**

**Rating:** 4
**Confidence:** 5

**Review:**

This paper attempts to address the semi-supervised learning topic by proposing a method based on an aggregated loss considering both cross-entry and Davies-Bouldin Index. Cross-entropy is used to ensure the maximum margin between classes and Davies-Bouldin Index is applied to the labeled data and to the whole dataset, respectively, to ensure a high quality of clustering. Evaluations in four small and simple datasets are reported to demonstrate the effectiveness of the proposed method.

The idea behind the proposed method is pretty simple and straightforward. It makes sense and so I believe that the method proposed is correct conceptually. However, a correct method may not necessarily be effective, which would require an extensive evaluation. That is what the paper lacks. Let me elaborate in detail.

First, all the four datasets used in the evaluations are small in scale and simple in class distributions. This is NOT the sate-of-the-art anymore. There are larger in scale and more complex datasets available and why did the authors fail to use these datasets instead?

Second, the comparison studies reported in Fig. 4 are not clearly documented and possibly are not fair either. Which specific supervised learning method did you use in this comparison? Did you use the same number of labeled data samples for the supervised learning method? The same questions are also raised for the toy example experiment reported in Fig. 3. If the supervised learning method uses the same number of the labeled samples, it is not a fair comparison obviously as your method uses more regularization constraints in the loss function. If a different number of the labeled samples is used, it is not a fair comparison either. Note that more constraints you use in the loss function, the number of labeled samples required can vary more, depending upon the different data distributions. Consequently, the comparison studies as well as the ablation studies reported in Fig. 4 can be misleading, possibly just showing the best case. More datasets with varying complexity in distributions may need to further the comparison studies before any meaningful conclusion can be reached.

Third, Davies-Bouldin Index is just one of the intrinsic evaluation metics for cluster analysis and there are others such as silhouette coefficient and Calinski-Harabasz score. In principle, you may also use these metics. I wonder whether you have investigated this and why you picked up Davies-Bouldin index.

Finally, the paper has a lot of grammatical errors and typos that should have been fixed before the paper was sent in for review.

As a minor comment, I don’t like the acronym MCMC for this proposed method. The same acronym is already in use as a very popular statistical machine learning method in the literature. The authors may want to change to another name.

---

### Official Review · AnonReviewer2 · 2020-10-28
**many weaknesses including relationship to existing work, lack of significant novelty, experiments, clarity**

**Rating:** 2
**Confidence:** 5

**Review:**

The proposed algorithm is not properly put in the context of other semi-supervised learning methods that are closely related. These include label propagation, clustering auto encoders, entropy minimization / margin maximization, prototype networks and semi-supervised clustering. Examples of these lines of research are:

Zhu and Ghahramani, Learning From Labeled and Unlabeled Data With Label Propagation,2002
Xie, Girschick and Farhadi, Unsupervised Deep Embedding for Clustering Analysis, 2016
Grandvalet and Bengio, Semi-supervised learning by entropy minimization, 2005
Snell et al, Prototypical Networks for Few-shot Learning, 2017

The idea of the paper is to penalize maximize distances between pairs of samples from different clusters as determined by the classifier output as a proxy for unlabeled samples. The paper claims previous works do not consider the spatial information provided by unlabeled examples. This is not correct. Many methods take advantage of such information such as label propagation (uses proximity between examples to propagate labels), entropy minimization (uses density of examples), etc. The novelty of the method over these previous techniques is not clear to me

The experiments section of the paper has some shortcomings as well. The datasets used in Figure 3 are also easily solved by other semi-supervised learning methods. The discussion of these figures does not lend insight into what type of problems that are challenging for previous methods are addressed by the proposed approach. Some of the experimental results do not appear to be compared to the state-of-the-art. For instance, Sajjadi et al, Regularization With Stochastic Transformations and Perturbations for Deep Semi-Supervised Learning, 2016 reports significantly better results for the MNIST dataset with 100 labeled examples compared to the results included in Table 1. Finally, the datasets used with the exception of MNIST are not standard ones used in recent semi-supervised learning papers. CIFAR-10 and STL-10 would have been better choices.

The clarity of the paper and English use is also below average. Some sentences contain informal phrases that might not be suitable for a scientific paper such as “People try…”, "Thanks for the tool; it helped us done differentiation automatically."  or vague statements such as "in some particular space, samples in the same category should be in the same cluster". The paper goes to lengths describing simple concepts such as cluster centroids and distance between cluster centroids (equations 1-3) yet concepts more crucial to the algorithm such as

---

### Official Review · AnonReviewer4 · 2020-10-28
**Interesting approach but weak representation**

**Rating:** 4
**Confidence:** 4

**Review:**

Summary:
The authors propose a novel loss function for semi-supervised learning. Arguing that SOTA semi-supervised learning methods neglect spatial information (latent clustering structure) in the data, the authors propose a loss function which combines clustering objectives with classification objectives. The proposed loss function combines the cross-entropy loss, the within-cluster scatter as known from k-means, the distance between centroids and the margin between classes. The proposed loss is notably non-continuous since it makes use of the maximum function. The authors employ  ADAM with a learning rate of 0.001 with exponential decay to optimize the novel loss function. Experiments on MNIST and three comparably small datasets show that the proposed method is able to achieve high accuracy with only few labeled data points.

Review:
Combining the elements of clustering and classification makes total sense for the case of semi-supervised learning. In that view, I find the conducted approach very interesting. However, the paper leaves many details unclear to the reader - in particular with regard to the definition of the loss function and the optimization (see minor issues below). I am missing a discussion about the optimization issues of the proposed loss function. The loss function is not continuous and in practice, this makes a loss susceptible to local minima. There might also be cases where numerical instabilities occur, for example if the centroids are close to each other and the DB loss is dividing by a small constant. In addition, I wonder about the sensitivity to the optimization parameters, such as the learning rate scheduling.
The experiments on synthetic data (half moons and three circles) look nice, but don't add much value. The classes/clusters in the datasets can be detected by simple clustering algorithms, additional information by a few labeled data points is not needed. Hence, these datasets are not suitable for demonstrating the power of semi-supervised methods. I would be more interested in examples where the points in one class are actually not connected/close to each other.
The experiments on real world data show a good performance of the proposed optimization of the novel loss function. Although  I don't believe that one always has to show experiments on a big, big dataset like imagenet, I think that at least a dataset of the magnitude of Cifar-10 and Cifar-100 should be considered. The learning rate is chosen to be comparatively small. How does that affect the convergence rate of the method on larger datasets? In addition, I would encourage a description of characteristics of the real world datasets which shows why they pose an interesting case (balanced/unbalanced classes, number of classes, easy/difficult to classify).

Minor issues:
* How are the centroids computed and when? If not all the data is labeled, then we can not compute the centroids as the average of points assigned to one cluster/class, because we need the centroid to determine the assignment of points to clusters/classes. Since DB is also reflecting the distance between centroids, the centroids are apparently additionally learned parameters? This aspect would become more clear if the loss functions would be denoted as functions. That is, instead of denoting a loss as, e.g., DB, denote it as DB(A,X). In Algorithm 1, L_{DB} is defined as DBI(Z_L,y). DBI is not defined as far as I see. I assume that DBI should actually reflect DB. Z are described as the network parameters, but what that entails is unclear.
* In deep learning, there is actually an inherent connection to k-means clustering. In fact, SOTA deep networks compute a latent embedding where every class is represented by a centroid. This is discussed here:
Hess, Sibylle, Wouter Duivesteijn, and Decebal Mocanu. "Softmax-based Classification is k-means Clustering: Formal Proof, Consequences for Adversarial Attacks, and Improvement through Centroid Based Tailoring." arXiv preprint arXiv:2001.01987 (2020).
This insight might be useful, since it says that the embedded space induced by cross-entropy learning *just* has to be tweaked in order to reflect a suitable clustering as well. This might lead to a more simple loss function. In addition, it puts the proposed approach more in perspective and adds theoretical value.
* What is the definition of k pairs single link distance in Eq.(5)?
* I wouldn't call the method MCMC, this is known as Markov Chain Monte Carlo.
* Table 1 returns the test error and Table 2 the test accuracy, use the same metric!
* Use the citation in the text, instead of saying "X et al. (X et al.) propose" say "X et al. propose" if the template allows it, where X et al. is the citation or say "(X et al.) propose".
* Define the parameters used in Algorithm 1.
* How many classes does the ETH dataset have? The description says only 8 categories with 10 subcategories.
* Please go over the text and use tools such as grammarly to improve the writing. For some senences I couldn't make out their meaning. For example: "Note that cross-entropy is is available o change into other losses in supervised learning for personal use."

---

### Official Review · AnonReviewer3 · 2020-10-29
**A paper with strong claims but lack of support and formalism in the technique**

**Rating:** 4
**Confidence:** 5

**Review:**

[Summary] In this work, authors dealt with the problem of improving classification performance in the setting of semi-supervised learning. Authors exploit inherent knowledge on the samples distribution via clustering to improve the latent space. The work heavily relies on well-studied concepts including the  Davies-Bouldin index and maximum margin clustering.


[Cons]
-- The general idea to improve the semi-supervised classification task by exploiting knowledge from the data through clustering constraints.


[Pros]
-- The lack of a formalist to describe the technique turns down the paper. This also makes unclear the level of novelty.
-- The results are ultimately less informative than one would like and the experimental setting is limited. This leads to the results being no more than case studies which demonstrates only limited advantages in both qualitative and quantitative terms.



Detailed comments for authors:


--  [Novelty] The novelty in the proposed technique is unclear as it leverages on already well-studied principles including Davies-Bouldin index and maximum margin clustering. Authors need to strongly  detail the main advantages and key insights of the technique. As it is described so far, it is hard to appreciate the level of contribution.

-- [Lack of formalism] A major drawback that turns down the paper is the lack of formalism to describe the proposed technique (which also makes unclear the level of novelty). That is, the core part of the technique is  (6) and previous expressions (1)-(5) are drawn from already well-studied concepts (Davies-Bouldin index and maximum margin clustering). One would expect that the authors provide further details (6) such as explicitly define the optimisation process.  Not only that, but several expressions are not properly set -- for example (5) authors needs to set the explicit definition rather that the text ‘k pairs single link distance’ as there are several works using the principles of maximum margin clustering, the reader needs to see the specifics of each component in the technique (even (5) draw from a very well-studied principle yet it is not defined properly). Overall, it is not enough to set (6) but authors should reduce (1)-(5) which are well-established expressions and detail (6) along with important parts such as the optimisation.

-- A major drawback of the components are not considered. For example, it is well-known that the Davies-Bouldin index can report good results, however, it does not imply the best information retrieval. Why then chose Davies-Bouldin index? There are several works such as [*] that can be used to generalise the latent space well whilst avoiding the disadvantage of the authors proposed components and affect positively the outcome from SSL [**].

[*]Caron, M., Bojanowski, P., Joulin, A., & Douze, M. (2018). Deep clustering for unsupervised learning of visual features. In Proceedings of the European Conference on Computer Vision (ECCV) (pp. 132-149).
[**] Sellars, Philip, Angelica Aviles-Rivero, and Carola Bibiane Schönlieb. "Two cycle learning: Clustering based regularisation for deep semi-supervised classification." arXiv preprint arXiv:2001.05317 (2020).


-- Experimental setting. The results are ultimately less informative than one would like and the experimental setting is limited. This leads to the results being no more than case studies which demonstrates only limited advantages in both qualitative and quantitative terms.
-- Authors open the paper with a big statement “the comparison results confirm our model’s outstanding performance over semi-supervised learning” However, authors use very small and not well-representative datasets such as CIFAR-100, ImageNet (or even a smaller version mini-ImageNet). Likewise the comparison setting is limited, there are no comparisons with recent other techniques such as FixMatch, UDA. Deep Label propagation etc. Otherwise the strong claims of the authors are unsupported.
-- The findings are less ultimate than one would like. The results from the tables are far from being well-explained.

---

### Decision · Program_Chairs · 2021-01-07
**Final Decision**

**Decision:**

Reject

**Comment:**

All reviewers agree that the writing is not precise. It does not help to find any novelty in the ideas, and the limited and too quickly described experiences are not convincing enough to forgive this problem. The authors chose not to oppose or comment on the detailed arguments provided by the reviewers. I agree with the reviewers in recommending the rejection of this paper.